# Prospective longitudinal study of 'Sleepless in Lockdown': unpacking differences in sleep loss during the coronavirus pandemic in the UK

Jane C Falkingham,[1] Maria Evandrou,[1,2] Min Qin [ID] ,[1] Athina Vlachantoni [ID] [1,2]

¹Faculty of Social Sciences, ESRC Centre for Population Change, University of Southampton, Southampton, UK
²Faculty of Social Sciences, Centre for Research on Ageing, University of Southampton, Southampton, UK

**Correspondence to**
Prof Jane C Falkingham;
J.C.Falkingham@soton.ac.uk

## ABSTRACT

**Objectives** COVID-19 is having a disproportionate impact on Black, Asian and minority ethnic (BAME) groups and women. Concern over direct and indirect effects may also impact on sleep. We explore the levels and social determinants of self-reported sleep loss among the UK population during the pandemic, focusing on ethnic and gender disparities.

**Setting** This prospective longitudinal study analysed data from seven waves of the Understanding Society: COVID-19 Study collected from April 2020 to January 2021 linked to prepandemic data from the 2019 mainstage interviews, providing baseline information about the respondents prior to the pandemic.

**Participants** The analytical sample included 8163 respondents aged 16 and above who took part in all seven waves with full information on sleep loss, defined as experiencing 'rather more' or 'much more' than usual sleep loss due to worry, providing 57 141 observations.

**Primary outcome measures** Self-reported sleep loss. Mixed-effects regression models were fitted to consider within-individual and between-individual differences.

**Results** Women were more likely to report sleep loss than men (OR 2.1, 95% CI 1.9 to 2.4) over the 10-month period. Being female, having young children, perceived financial difficulties and COVID-19 symptoms were all predictive of sleep loss. Once these covariates were controlled for, the bivariate relationship between ethnicity and sleep loss (1.4, 95% CI 1.6 to 2.4) was reversed (0.7, 95% CI 0.5 to 0.8). Moreover, the strength of the association between gender and ethnicity and the risk of sleep loss varied over time, being weaker among women in July (0.6, 95% CI 0.5 to 0.7), September (0.7, 95% CI 0.6 to 0.8), November (0.8, 95% CI 0.7 to 1.0) and January 2021 (0.8, 95% CI 0.7 to 0.9) compared with April 2020, but positively stronger among BAME individuals in May (1.4, 95% CI 1.0 to 2.1), weaker only in September (0.7, 95% CI 0.5 to 1.0).

**Conclusions** The pandemic has widened sleep deprivation disparities, with women with young children, COVID-19 infection and BAME individuals experiencing sleep loss, which may adversely affect their mental and physical health.

## INTRODUCTION

The coronavirus disease (COVID-19) is impacting on physical and mental health

### Strengths and limitations of this study

► This is a large-scale community-based prospective longitudinal study.
► We examined the trends and patterns of sleep problems and the impact of known correlates of being infected with, or affected by, COVID-19 including ethnicity and gender.
► Sleep loss was measured by self-reports. As no definition of sleep loss was provided, participants may have used different definitions while answering the question.
► Attrition could have confounded the interested prospective associations.

globally. Sleep problems associated with increased psychosocial stressors induced by the coronavirus itself, and as a result of the social distancing measures that have been introduced to manage the virus, are emerging as a significant outcome of the COVID-19 crisis. According to a report last summer, more than half of the UK population has struggled with sleep during the first lockdown.[1] Sleep has long been recognised as an essential determinant of human health and performance. Good sleep restores energy, promotes healing, interacts with the immune system and impacts on behaviour.[2] Even acute sleep deprivation can impair judgement and cognitive performance, while persistent deviations have been linked to disease development and increased mortality.[3 4] During the pandemic, lack of sleep may itself have had knock-on effects on people's capacity to be resilient. However, to date, relatively limited research has been conducted on sleep deprivation during the pandemic.

Sleep is known to be regulated by circadian rhythms, sleep-wake homoeostasis and cognitive–behavioural influences.[3] With regard to environmental, behavioural and health determinants, poor sleep has been associated

**Table 1** Analytical sample characteristics (wave 1)

| | Analytical sample | Full wave 1 sample | P value* |
|---|---|---|---|
| Age, mean | 50.6 (SD=17.5) (8163) | 50.1 (SD=18.2) (17 452) | 0.036 |
| Age group, % (n) | | | <0.001 |
| 16–24 | 9.2 (309) | 10.2 (1543) | |
| 25–44 | 27.3 (1587) | 28.6 (4734) | |
| 45–64 | 38.9 (3571) | 36.9 (7028) | |
| 65–74 | 16.6 (1972) | 15.0 (2925) | |
| 75+ | 8.0 (724) | 9.4 (1222) | |
| Gender, % (n) | | | 0.092 |
| Men | 47.0 (3418) | 48.0 (7287) | |
| Women | 53.0 (4745) | 52.0 (10 165) | |
| Ethnicity, % (n) | | | <0.001 |
| British/English/Scottish/Welsh/Northern Irish (White) | 89.8 (7138) | 86.5 (14 029) | |
| Other White | 3.3 (343) | 3.7 (779) | |
| Black, Asian and minority ethnic (BAME) | 6.1 (577) | 8.6 (2044) | |
| Highest qualification, % (n) | | | 0.001 |
| No qualification | 5.1 (317) | 6.3 (650) | |
| General Certificate of Secondary Education (GCSE) or lower | 28.3 (1994) | 29.9 (4006) | |
| A level | 22.8 (1599) | 22.4 (3401) | |
| Degree | 43.8 (4210) | 41.4 (8195) | |
| Live with a partner, % (n) | | | 0.041 |
| No | 35.3 (2114) | 36.1 (5136) | |
| Yes | 64.7 (6049) | 63.9 (12 316) | |
| Children in the house, % (n) | | | <0.001 |
| At least one child aged 0–4 | 8.2 (552) | 9.0 (1756) | |
| At least one schoolchild aged 5–18 | 22.0 (1488) | 23.5 (4410) | |
| No schoolchildren | 69.8 (6123) | 67.4 (11 286) | |
| Key worker, % (n) | | | <0.001 |
| No | 33.2 (2479) | 30.8 (5588) | |
| Yes | 27.1 (1997) | 25.1 (4515) | |
| Not in paid or self-employed work | 39.6 (3687) | 44.1 (7349) | |
| Has had symptoms that could be coronavirus, % (n) | | | 0.017 |
| No | 89.1 (7266) | 88.2 (15 250) | |
| Yes | 10.9 (897) | 11.8 (1305) | |
| Feel lonely, % (n) | | | 0.044 |
| Hardly ever | 62.3 (5488) | 59.1 (10 717) | |
| Sometimes | 28.9 (2169) | 29.3 (4947) | |
| Often | 8.8 (506) | 8.4 (1276) | |
| Subjective current financial situation, % (n) | | | <0.001 |
| Living comfortably | 32.8 (3261) | 29.6 (5815) | |
| Doing alright | 42.2 (3441) | 40.1 (7045) | |
| Just about getting by | 19.6 (1122) | 17.2 (2585) | |
| Finding it quite difficult/very difficult | 5.4 (339) | 6.2 (891) | |
| Subjective future financial situation, % (n) | | | <0.001 |
| Better off | 6.5 (523) | 7.1 (1248) | |

**Table 1** Continued

| | Analytical sample | Full wave 1 sample | P value* |
|---|---|---|---|
| Worse off | 16.0 (1245) | 15.3 (2793) | |
| About the same | 77.4 (6395) | 70.7 (12 278) | |
| Prior problem of sleep, % (n) | | | 0.049 |
| No | 83.9 (7031) | 78.5 (13 668) | |
| Yes | 16.1 (1132) | 15.8 (2538) | |
| Sleep loss, % (n) | | | 0.001 |
| No | 76.8 (6371) | 75.2 (12 008) | |
| Yes | 23.2 (1792) | 24.8 (3919) | |
| Region, % (n) | | | <0.001 |
| North East | 4.8 (291) | 4.2 (593) | |
| North West | 9.9 (774) | 10.9 (1716) | |
| Yorkshire and the Humber | 9.1 (712) | 8.9 (1482) | |
| East Midlands | 8.7 (659) | 7.7 (1334) | |
| West Midlands | 9.1 (669) | 9.1 (1479) | |
| East of England | 9.8 (805) | 10.0 (1689) | |
| London | 9.1 (658) | 11.7 (1849) | |
| South East | 15.2 (1208) | 14.1 (2428) | |
| South West | 10.6 (880) | 9.0 (1598) | |
| Wales | 4.3 (454) | 4.5 (1018) | |
| Scotland | 7.0 (728) | 7.6 (1523) | |
| Northern Ireland | 2.5 (325) | 2.3 (742) | |

Source: Authors' analysis, Understanding Society: COVID-19 Study, 2020.

Proportions for the analytical sample are weighted using longitudinal weight; proportions for the entire sample at wave 1 are weighted using cross-sectional weight. The number of respondents is unweighted. The mean age difference test used the t-test. Other categorical variables used the Pearson $\chi^2$ test.

*P value for comparison between participants in the analytical sample and all in the wave 1.

with stress, anxiety, work pressures, financial concerns, mental and physical impairments, and physical activity.[5–8] Previous studies have found that women were more likely than men to have trouble falling and staying asleep frequently, or to have insufficient sleep.[9 10] The relationship between ethnicity and sleep is complicated due to the broader social and environmental factors determining group differences in sleep behaviours and the structural relationships between these factors and ethnicity.[11] Some studies reported that inadequate sleep duration and poorer sleep were more prevalent among low-income and Black, Asian and minority ethnic (BAME) communities,[12] whereas others have failed to find this association.[13]

In understanding the relationship between COVID-19 and sleep, it is helpful to conceptually distinguish between those factors linked to being *infected* with COVID-19 and those associated with the policy responses and measures introduced to manage the pandemic that have *affected* everyday life. Although it is still relatively early in our understanding of COVID-19, research by Public Health England 2020 has highlighted that older people, men and individuals from BAME groups are all at increased risk of developing a severe response to the virus and to die from it.[14] The reasons for the heightened risk among certain ethnic groups remain unresolved, but potential contributors include the disproportionate representation of BAME individuals in some high-risk occupations including front-line healthcare work, as well as wider environmental factors that, interwoven with issues of inequality, deprivation and structural racism, manifest in long-standing ethnic disparities in health.[15] A priori, we might expect those groups facing the greatest health risks from the virus to report increased sleep loss due to worry and thus to observe differences across ethnic groups.

The public health actions taken to control the spread of the virus have, however, impacted all domains of life and thus affected all individuals. On 23 March 2020, the UK went into lockdown in an unprecedented attempt to limit the spread of coronavirus, with the government mandating all those who could to work at home, closing schools, restaurants and all but essential shops, and advising the population to stay at home and limit contact with other individuals outside their household. On 5 November, a second national lockdown came into force in England, followed by the third nationwide lockdown on 5 January 2021, the phased end of which is scheduled

for 21 June. The resultant move to home working and learning and, for some, the loss of work altogether, along with limited social contact and increased isolation, may all be anticipated to affect mental well-being and the ability to sleep. Preliminary evidence points towards the young and women being disproportionally affected, with women being more likely than men to be working in sectors that were locked down[16] and mothers being more likely to be interrupted while working from home than fathers.[17] Lockdown has also resulted in increased instances of domestic violence; the UK domestic abuse organisation, Refuge, reported a 25% increase in calls and online requests since the first lockdown began in March 2020.[18] Given this, we might anticipate a gender differential in increased sleep loss, with women being disproportionately affected by lockdown compared with men.

This study aims to provide novel evidence regarding patterns of self-reported increased sleep loss due to worry during the first 4 weeks of the COVID-19-related lockdown in the UK. Using recently collected nationally representative survey data, the research provides the first estimates of the prevalence and incidence of increased sleep loss since the coronavirus pandemic. It attempts to unpack the impact of factors associated with being *infected* and being *affected* by COVID-19, with a particular focus on the extent to which the pandemic has exacerbated differentials in sleep loss by ethnicity and gender.

## METHODS
### Study design and population
We analysed data from the Understanding Society (USoc): COVID-19 Study[19] covering 10 months and three COVID-19-related national lockdowns in the UK. Data were collected online monthly. The first wave of the COVID-19 survey was fielded between 24 and 30 April 2020, with wave 2 taking place in May, wave 3 in June, wave 4 in July, wave 5 in September, wave 6 in November and wave 7 in December 2020/January 2021. There was no survey in August or October. The UK Household Longitudinal Study[20] (UKHLS; USoc) is an ongoing panel survey of more than 40 000 households that began in 2009. Between 24 and 30 April 2020, members of households who participated in either of the two most recent UKHLS data collections (wave 8 or 9), and who were older than 16 years, were invited to complete the first wave of the COVID-19 web survey. The probability sample was drawn from postal addresses. Northern Ireland and areas in England, Scotland and Wales with proportionately large migrant and ethnic minority populations were oversampled. All household members aged 16 or older in April 2020 were invited to participate, except for those unable to make an informed decision as a result of incapacity, and those with unknown postal addresses or addresses abroad. The response rate (full interview) of seven waves of the USoc: COVID-19 Study was 39%, 35%, 33%, 32%, 30%, 28% and 28%, respectively.[19]

The USoc: COVID-19 Study is funded by the Economic and Social Research Council and the Health Foundation with scientific leadership by the Institute for Social and Economic Research, University of Essex. Fieldwork for the survey is carried out by Ipsos MORI and Kantar. The research data are distributed by the UK Data Service.

There were 17 452 respondents who took part in wave 1 of the COVID-19 study. The inclusion criteria of the analytical sample for this study were all respondents aged 16 and over, who had participated in all seven waves survey and had no missing values on the outcome variable, constituting a final sample size of 8163 individuals and 57 141 observations. The characteristics of the analytical sample are shown in table 1. Compared with all participants in wave 1, the analytical sample was slightly older and wealthier than the baseline sample. BAME individuals were less likely to continue to participate in the follow-up studies. Loss to follow-up was slightly more likely among those who reported sleep loss at wave 1 of the COVID-19 study.

### Procedures
The outcome variables included whether the respondent reported an increase in sleep loss over worry in the last few weeks. The question on sleep loss was identical across both the USoc: COVID-19 Study and the mainstage data in 2019. As participants were not provided with a definition of sleep loss, it is recognised that different participants may have used different definitions when answering the question. The specific question wording, along with the four response categories, is presented in the text box below.

---

**The next questions are about how you have been feeling over the last few weeks.**
Have you recently lost much sleep over worry?
1. Not at all.
2. No more than usual.
3. Rather more than usual.
4. Much more than usual.
*Source*: University of Essex (2021).

---

In addition to the general sleep loss question, in the wave 4 COVID-19 study, participants were asked a more detailed set of questions about sleep, including: How many hours of actual sleep did you usually get per night during the last month? How often have you had trouble sleeping because you cannot get to sleep within 30 min? How often have you had trouble sleeping because you wake up in the middle of the night or early in the morning? These more comprehensive measures were compared against the general sleep loss measure as part of the descriptive analyses.

For the purposes of the multivariate analysis, a respondent was defined as experiencing sleep loss during the lockdown if he or she reported sleep loss over worry in the last few weeks 'rather more than usual' or 'much more than usual' in the USoc: COVID-19 Study. The outcome variables were binary (1=yes; 0=no).

A range of explanatory variables were included, reflecting both known associates of sleep loss[9 21] as well as those that we hypothesise may be associated with heightened anxiety during the pandemic. Demographic and socioeconomic characteristics included age, gender, ethnicity and educational qualification. Gender distinguished between men and women; the survey responses do not differentiate those whose reported sex has changed since birth or those who classify themselves as intersex. Ethnicity was classified into three groups: British/English/Scottish/Welsh/Northern Irish (White), Other White, and BAME.

Variables capturing factors associated with COVID-19 itself included whether the respondent reported having experienced symptoms that could be coronavirus and being a key worker. (According to Department of Health and Social Care guidance on testing eligibility,[22] key workers are people whose jobs are vital to public health and safety during the coronavirus lockdown. The list includes health and social care, for example, all National Health Service (NHS) staff, front-line health and social care staff such as doctors, nurses, plus support and specialist staff in the health and social care sector; education and childcare, including social workers; food and other necessary goods; key public services; local and national governments; utility workers; public safety and national security; and transport.)[22] Other variables aimed to capture the impacts of the policy response to COVID-19, particularly the effect of lockdown. Increased stress related to childcare and home schooling was proxied by the presence of children in the house (whether at least one child aged 0–4 or school-age child), and whether the respondent was living with a partner. Exposure to financial stress was proxied by two variables capturing the respondents' subjective view of their current and future financial situation. Social isolation was measured by the frequency of having felt lonely in the last 4 weeks. In order to capture regional differences in the intensity of the pandemic (In the first few weeks of the pandemic in the UK, the rate of infection was much higher in London than elsewhere in the country), a variable reflecting the respondents' place of respondence was included based on the UK government office region (North East, North West, Yorkshire and the Humber, East Midlands, West Midlands, East of England, London, South East, South West, Wales, Scotland and Northern Ireland). Finally, in order to capture time effects, we controlled for the survey time point, measured as a dummy variable (April, May, June, July, September, November and December/January).

Sleep loss and certain covariates such as the frequency of feelings of loneliness in the previous 4 weeks, has had symptoms that could be coronavirus, young children present in the household and the respondents' perceived current and future financial situation were measured at each wave of COVID-19 study. For some individuals, their responses to these questions changed over time during the follow-up period; thus, in the statistical models, these were all treated as time-varying variables. Other covariates were treated as time-invariant variables. It is recognised that pregnancy, menstrual cycle and menopause may all be linked to sleep. Data on these were not available. However, age was included as a covariant, which will partially capture whether women have reached the menopause.

## Analytical approach

Descriptive statistics illustrating the prevalence of reported sleep loss (stratified by gender and ethnicity) before and during the pandemic among *all* participants aged 16 and above in the USoc mainstage survey in 2019 and in the COVID-19 study waves 1–7 were calculated (figure 1).

Subsequent analysis was then restricted to the analytical sample described above. Mixed-effects logistic models were used to assess the existence and strength of associations between sleep loss and COVID-19-related circumstances during the pandemic. Given that the data collected from an individual over 10 months are not independent of each other, mixed-effects models are the recommended statistical technique to take into consideration between-individuals variance and within-individuals variance.[23] This method has the advantage of including both fixed and random effects. The former are model components used to define systematic relationships such as overall changes of sleep loss over time and/or social determinants induced by individual differences, and the latter account for the variability among individuals around the systematic relationships captured by the fixed effects.

For this study, the logistic mixed-effects models included sleep loss as the response variable, and gender, ethnicity group and time point as fixed effects. Observations by person-month were attached to each respondent; thus, in order to estimate the random intercepts, we accounted for random variation between individuals and between observations within the same individual. We modelled an interaction between time and gender, as well as between time and ethnicity, as a fixed effect to examine whether the association between gender and ethnicity varied according to time. Other potential confounding variables included the respondents' age, highest qualification, subjective current financial situation, subjective future financial situation, whether they were living with a partner, any children in the house, whether they were a key worker, the frequency of feelings of loneliness in the previous 4 weeks, any prior problem of sleep and region. Regression analyses were carried out for the total population and then for men and women separately. Results were presented as ORs of sleep loss by gender and ethnicity, with 95% CIs and associated two-sided p values.

A sensitivity analysis was conducted in order to check the robustness of the results. Here we measured sleep loss as an ordered variable with four categories (not at all, no more than usual, rather more than usual and much more than usual) instead of two. Mixed-effects ordered logistic

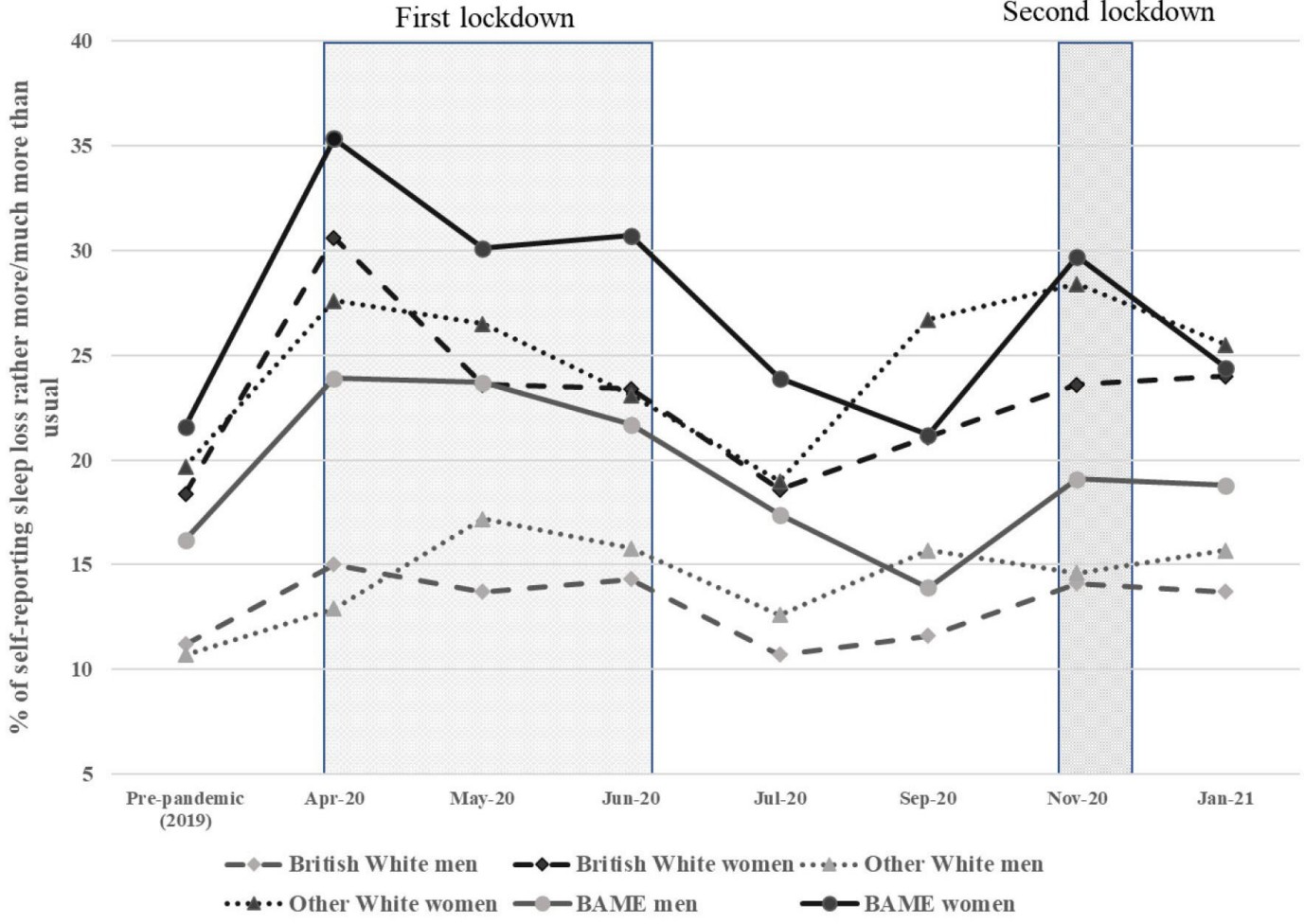

**Figure 1** Prevalence of reported sleep loss before and during the pandemic among all participants aged 16 and above in Understanding Society mainstage survey in 2019 and Understanding Society: COVID-19 Study wave 1 to wave 7. Source: Authors' analysis, Understanding Society: COVID-19 Study, 2020. Number of respondents: n=29 685 (prepandemic 2019), n=15 668 (April 2020), n=14 154 (May 2020), n=13 437 (June 2020), n=13 075 (July 2020), n=12 170 (September 2020), n=11 472 (November 2020), n=11 299 (January 2021). All proportions are unweighted. Black lines represent women in different ethnicities and grey lines represent men. BAME, black, Asian and minority ethnic.

regression was applied. The analyses were carried out in STATA V.15.[24]

### Patient involvement
No patients were involved in this study.

### RESULTS
#### Descriptive analyses
Figure 1 shows the level of reported sleep loss by gender and ethnicity among all participants aged 16 and above in the USoc mainstage survey in 2019 and in the USoc COVID-19 Study wave 1 to wave 7. The first data point provides a baseline, prepandemic, with subsequent data points then illustrating how sleep loss changed during the pandemic, with the first two national lockdowns highlighted, the third starting in January 2021. Black lines represent women in different ethnicities and grey lines represent men. Overall, women reported much higher levels of sleep loss than men; and respondents from

ethnic minority groups had a higher prevalence of sleep loss than British White respondents among both men and women, with BAME women experiencing the highest prevalence. Differentials in the prevalence of reporting sleep loss by gender and ethnicity are evident at baseline; however, the gap appears to have widened during the pandemic and was especially pronounced in April 2020, illustrating the effect of the early stages of the pandemic and the first lockdown, when the prevalence of sleep loss rose among all groups, with the rise particularly marked among all women and BAME men. As the first lockdown eased, the prevalence of sleep loss fell, reaching a low in July/September, although for most groups this remained slightly above the prepandemic baseline. Sleep loss then increased during the autumn, with a second peak coinciding with the second national lockdown in November 2020. The prevalence of reported sleep loss before and during the pandemic among the analytical sample (online supplemental appendix figure 1) shows a similar pattern.

However, sleep loss among BAME women was lower in the analytical sample than among all respondents without follow-up selection, while the level of sleep loss among 'Other White' women was higher.

Restricting the analytical sample to all those with complete information across all seven waves, table 2 shows that one in four people in the UK reported increased sleep loss due to worry during the first 4 weeks of the first coronavirus pandemic lockdown in spring 2020 (table 2). There were clear differences between women and men and across ethnic communities. Strikingly, although men have been found to face a higher risk of experiencing severe symptoms and dying from COVID-19, women were twice as likely as men to report that they had lost sleep 'much more than usual' (5.8% vs 1.9%) and 'rather more than usual' (23.9% vs 11.8%), supporting the hypothesis that women were disproportionately affected by the economic and social consequences of the first lockdown. In the consecutive months, the proportion reporting increased sleep loss slightly decreased among both men and women, but women still showed a much higher proportion of increased sleep loss than men.

The results by ethnicity do, however, support the argument that the risk of infection may play a role; in the first month of lockdown in spring 2020, 6.3% of BAME respondents reported being 'much more than usual' to have lost sleep through worry over the last few weeks compared with 3.7% of white respondents (table 2). Patterns of the changes of per cent across waves are illustrated using stacked bar charts for gender and ethnicity in figure 2A,B.

The prevalence of sleep loss ('rather more than usual' and 'much more than usual') in April 2020 (22.0%) was higher than that reported prior to the pandemic (13.9%) and the differentials between men and women and individuals from different ethnic communities have widened during the epidemic (online supplemental appendix table A).

Looking at the more detailed information available in wave 4 (July 2020) shows that on average, those who reported problems with sleep slept around 45 min less than those who did not have trouble sleeping (table 3). The majority of respondents reporting sleep loss had trouble getting to sleep within 30 min or reported waking up in the middle of the night or early in the morning. They were also more likely to rate their overall quality of sleep fairly bad or very bad. They were also more likely to report having trouble sleeping because of having a cough or snoring loudly, taking medicine to help them sleep and having trouble staying awake while driving, eating meals or engaging in social activity (table 3), indicating that the general sleep loss question is a good proxy for this wider set of sleep-related issues.

## Multivariate analysis

Many of the characteristics associated with sleep loss are likely to be inter-related; for example, those with a child aged 0–4 at home are also likely to be aged 25–44; those

who are key workers are more likely to have experienced symptoms, etc. To further unravel the picture, a series of multivariate logistic regression models were run; first, for the population as a whole, and then separately for men and women. The first two columns (A1 and A2) for model A (table 4) show the unadjusted (bivariate) ORs of experiencing sleep loss by gender and ethnicity (OR=3.2, 95% CI 2.8 to 3.6; OR=1.4, 95% CI 1.6 to 2.4), while the third column (A3) shows the adjusted odds of the main effect for the full model.

The analysis shows that the differential reporting of sleep loss by gender remains significant even after controlling for all other factors (OR=2.1, 95% CI 1.9 to 2.4); the differential by ethnicity is however reversed (OR=0.7, 95% CI 0.5 to 0.8) once other factors are taken into account such as being a key worker, having had symptoms, having children in the household, experiencing financial difficulties and living in London, the initial epicentre of the UK's COVID-19 outbreak in spring 2020. Individuals from BAME communities are disproportionately represented in all of these groups (A range of interaction effects with ethnicity were investigated but none were significant and thus are not included in the final model) (online supplemental appendix table B). By including interactions between gender and ethnicity with month, model A4 assessed the effects of gender and ethnicity on shifts over time. The results show that the strength of the association between gender and ethnicity and the risk of sleep loss varied over time. Women presented a lower risk of sleep loss in May, June, July, November and January 2021 as compared with the early stages of the first lockdown in April 2020 (OR=0.7, 95% CI 0.5 to 0.8; OR=0.6, 95% CI 0.5 to 0.8; OR=0.7, 95% CI 0.6 to 0.9; OR=0.8, 95% CI 0.6 to 0.9; OR=0.7, 95% CI 0.6 to 0.9). By contrast, the BAME group experienced a higher risk of sleep loss in May (OR=1.4, 95% CI 1.0 to 2.1), and the risk only started to fall in September (0.7, 95% CI 0.5 to 1.0).

The analysis shows that the coronavirus infection, with school-age children at home, feeling lonely, perceived financial difficulties and worry, being a woman and month were all predictive factors of sleep loss (table 4). The influential factors were slightly different among men (model B) and women (model C). Among women, experiencing coronavirus symptoms was a risk factor while being of BAME heritage reduced the risk of increased sleep loss *once other factors were controlled for*, while both these factors were not significant for men.

In the sensitivity analysis where sleep loss was measured as an ordered variable, the analysis of the data with mixed-effects ordered logistic regression rather than mixed-effects logistic regression did not change the pattern of results (online supplemental appendix table C).

## DISCUSSION AND CONCLUSION

This study has revealed several important findings related to sleep health during the COVID-19 pandemic. First, it provides robust evidence that sleep loss is affecting more

**Table 2** Prevalence of reported sleep loss at each wave of the Understanding Society: COVID-19 Study (n=8163)

| | | Not at all | No more than usual | Rather more than usual | Much more than usual |
|---|---|---|---|---|---|
| Wave 1 (April 2020) | All participants | 35.3 | 42.4 | 18.3 | 4.0 |
| | By gender | P<0.001 | | | |
| | Men | 45.3 | 41.0 | 11.8 | 1.9 |
| | Women | 26.7 | 43.6 | 23.9 | 5.8 |
| | By ethnicity | P<0.001 | | | |
| | British/English/Scottish/Welsh/Northern Irish (White) | 35.5 | 42.3 | 18.5 | 3.7 |
| | Other White | 34.2 | 42.6 | 16.8 | 6.4 |
| | Black, Asian and minority ethnic (BAME) | 18.5 | 16.8 | 16.7 | 6.3 |
| Wave 2 (May 2020) | All participants | 33.3 | 47.5 | 15.9 | 3.3 |
| | By gender | P<0.001 | | | |
| | Men | 42.6 | 43.4 | 11.8 | 2.2 |
| | Women | 24.9 | 51.2 | 19.7 | 4.2 |
| | By ethnicity | P<0.001 | | | |
| | British/English/Scottish/Welsh/Northern Irish (White) | 33.8 | 47.6 | 15.6 | 3.0 |
| | Other White | 30.2 | 40.6 | 23.8 | 5.4 |
| | BAME | 27.8 | 50.6 | 15.3 | 6.3 |
| Wave 3 (June 2020) | All participants | 30.1 | 49.6 | 16.5 | 3.8 |
| | By gender | P<0.001 | | | |
| | Men | 36.9 | 47.0 | 13.1 | 3.0 |
| | Women | 24.1 | 51.9 | 19.6 | 4.4 |
| | By ethnicity | P<0.001 | | | |
| | British/English/Scottish/Welsh/Northern Irish (White) | 30.7 | 49.4 | 16.4 | 3.5 |
| | Other White | 27.3 | 44.4 | 21.2 | 7.1 |
| | BAME | 24.3 | 55.2 | 13.9 | 6.5 |
| Wave 4 (July 2020) | All participants | 35.4 | 48.5 | 13.6 | 2.5 |
| | By gender | P<0.001 | | | |
| | Men | 44.9 | 43.6 | 10.2 | 1.3 |
| | Women | 26.9 | 52.8 | 16.6 | 3.7 |
| | By ethnicity | P=0.031 | | | |
| | British/English/Scottish/Welsh/Northern Irish (White) | 35.8 | 48.4 | 13.3 | 2.5 |
| | Other White | 30.1 | 53.1 | 15.3 | 1.5 |
| | BAME | 34.0 | 46.0 | 15.9 | 4.2 |
| Wave 5 (September 2020) | All participants | 31.0 | 52.3 | 13.8 | 2.9 |
| | By gender | P<0.001 | | | |
| | Men | 38.4 | 50.4 | 9.6 | 1.6 |
| | Women | 24.5 | 54.0 | 17.4 | 4.0 |
| | By ethnicity | P=0.004 | | | |

**Table 2** Continued

| | | Not at all | No more than usual | Rather more than usual | Much more than usual |
|---|---|---|---|---|---|
| | British/English/Scottish/Welsh/ Northern Irish (White) | 31.3 | 52.1 | 13.7 | 2.9 |
| | Other White | 28.8 | 49.3 | 19.0 | 2.9 |
| | BAME | 31.5 | 53.6 | 12.5 | 2.3 |
| Wave 6 (November 2020) | All participants | 26.2 | 53.3 | 16.9 | 3.7 |
| | By gender | P<0.001 | | | |
| | Men | 33.6 | 51.4 | 12.8 | 2.2 |
| | Women | 19.7 | 54.9 | 20.5 | 5.0 |
| | By ethnicity | P=0.03 | | | |
| | British/English/Scottish/Welsh/ Northern Irish (White) | 26.5 | 53.2 | 16.7 | 3.5 |
| | Other White | 22.5 | 53.2 | 20.6 | 3.7 |
| | BAME | 23.8 | 52.1 | 18.9 | 5.2 |
| Wave 7 (January 2021) | All participants | 27.5 | 53.0 | 15.9 | 3.7 |
| | By gender | P<0.001 | | | |
| | Men | 34.6 | 51.7 | 10.7 | 3.0 |
| | Women | 21.1 | 54.1 | 20.6 | 4.3 |
| | By ethnicity | P<0.001 | | | |
| | British/English/Scottish/Welsh/ Northern Irish (White) | 28.2 | 52.3 | 16.1 | 3.5 |
| | Other White | 23.6 | 50.5 | 21.6 | 4.8 |
| | BAME | 20.1 | 64.2 | 11.0 | 4.8 |

Source: Authors' analysis, Understanding Society: COVID-19 Study, 2020.
All proportions are weighted using each wave cross-sectional individual web survey weight, beta version. The number of respondents is unweighted. Pearson $\chi^2$ test applied.

people during the COVID-19 pandemic than previously, reflecting the fact that stress levels have risen due to anxieties about health, financial consequences, changes in social life and the daily routine, all of which may affect sleep homeostasis.

The study also provides evidence that women have been more vulnerable to sleep deprivation during lockdown, which is in line with previous research suggesting that women have more sleep disturbances than men,[9 21] and that women are more prone to stress-related sleep disorders such as post-traumatic stress disorder and anxiety disorders.[25] There is emerging evidence that mental health experiences during the COVID-19 pandemic in the UK differ between men and women, with more women suffering from anxiety in the early stages of lockdown.[26] Women's position in the labour market may increase their exposure to COVID-19, as women represent a significant majority of front-line workers in social care, education and healthcare.[16 27] Many parents will have been affected by school closures, and requirements to balance paid

work with increasing childcare and providing support to their children's learning. However, the gendered allocation of childcare means that in many households, it is the mother who has continued to provide the majority of primary care for children. Furthermore, many mid-life women find themselves juggling employment with caring responsibilities for aged parents and grandchildren.[28]

Individuals from BAME communities showed a higher prevalence and incidence rate of sleep loss than British White individuals. This reflects the fact that those of BAME heritage have disproportionally higher rates of coronavirus infection,[14] high anxiety associated with coronavirus-specific circumstances, are more likely to be key workers, to have dependent children and to feel lonely. All of these factors are likely to increase the risk of sleep loss, with the result that once all these other factors are controlled for, being a member of a BAME community was associated with a *reduced* chance of sleep loss—highlighting the complex relationship between ethnicity and sleep health. One possible hypothesised explanation

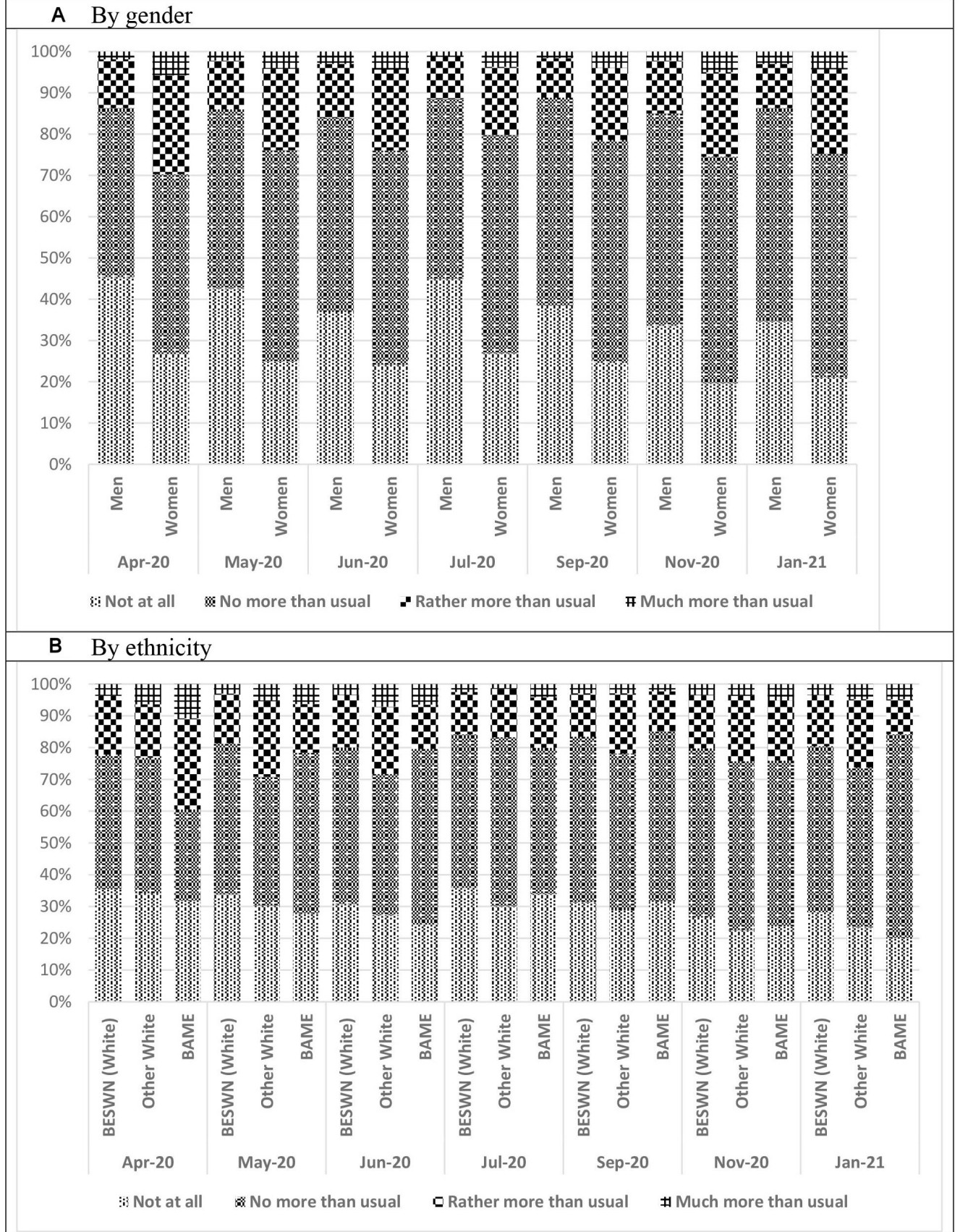

**Figure 2** Prevalence of reported sleep loss at each wave of the Understanding Society: COVID-19 Study (n=8163) by Gender (A) and Ethnicity (B). Source: Authors' analysis, Understanding Society: COVID-19 Study, 2020. All proportions are weighted using each wave cross-sectional individual web survey weight, beta version. BAME, black, Asian and minority ethnic; BESWN, British/English/Scottish/Welsh/Northern Irish.

**Table 3** Self-reported sleep loss and its association with more detailed information on trouble sleeping, July 2020

| | No sleep loss problem | Sleep loss problem | P value |
|---|---|---|---|
| Had trouble sleeping because I cannot get to sleep within 30 min | | | <0.001 |
| Not during the past month | 38.3 | 8.0 | |
| Less than once a week | 27.7 | 10.9 | |
| Once or twice a week | 20.6 | 25.3 | |
| Three or more times a week | 7.5 | 29.0 | |
| More than once most nights | 5.8 | 24.9 | |
| Had trouble sleeping because I wake up in the middle of the night or early in the morning | | | <0.001 |
| Not during the past month | 28.5 | 2.2 | |
| Less than once a week | 23.9 | 8.8 | |
| Once or twice a week | 22.9 | 28.6 | |
| Three or more times a week | 13.4 | 31.4 | |
| More than once most nights | 10.8 | 28.8 | |
| Had trouble sleeping because I cough or snore loudly | | | <0.001 |
| Not during the past month | 83.9 | 71.4 | |
| Less than once a week | 8.7 | 10.5 | |
| Once or twice a week | 4.2 | 9.5 | |
| Three or more times a week | 1.5 | 4.1 | |
| More than once most nights | 1.3 | 2.6 | |
| Taken medicine to help sleep | | | <0.001 |
| Not during the past month | 93.3 | 78.0 | |
| Less than once a week | 2.0 | 5.1 | |
| Once or twice a week | 1.8 | 5.1 | |
| Three or more times a week | 2.7 | 9.6 | |
| Had trouble staying awake while driving, eating meals or engaging in social activity | | | <0.001 |
| Not during the past month | 89.7 | 71.8 | |
| Less than once a week | 6.0 | 13.3 | |
| Once or twice a week | 3.0 | 10.3 | |
| Three or more times a week | 1.1 | 2.5 | |
| Overall quality of sleep | | | <0.001 |
| Very good | 20.1 | 0.6 | |
| Fairly good | 62.1 | 31.2 | |
| Fairly bad | 15.9 | 52.9 | |
| Very bad | 1.9 | 15.2 | |
| Average sleep hours | 7.09 (SD=1.22) | 6.35 (SD=1.20) | <0.001 |
| Respondents (n) | 6974 | 1189 | |

Source: Authors' analysis, Understanding Society: COVID-19 Study, 2020. Wave 4.
All proportions are weighted using wave 4 cross-sectional individual web survey weight, beta version. The number of respondents is unweighted. Average sleep hours difference test used analysis of variance (ANOVA) F test, others used Pearson $\chi^2$ test.

might be that there are some differences in sleep patterns between the different ethnicities included in the BAME definition, given the vast social and cultural heterogeneity within the BAME population. Sleep is culturally practised across racial and ethnic groups.[29] For example, some cultures in Latin and Caribbean communities promote taking naps in the middle of the day (siesta) and biphasic sleep. Others endorse the polyphasic sleep modality in Asia and Africa, anchoring their sleep at night but taking several daytime naps as needed when under a social condition that allows sleep to occur.[29] A review study[30] shows that among the few published within-racial/ethnic group analyses, there are differences in sleep between non-US-born and US-born racial/ethnic groups. Still, the group with the more favourable sleep profile is consistent for non-US-born Latinos compared with US-born Latinos

**Table 4** Multivariate modelling of the correlates of sleep loss during the pandemic (ORs and 95% CI)

| | Model A All respondents | | | | Model B Men | | Model C Women | |
|---|---|---|---|---|---|---|---|---|
| | A1 | A2 | A3 | A4 | B1 | B2 | C1 | C2 |
| Gender | | | | | | | | |
| Men (ref) | | | | | | | | |
| Women | 3.2*** (2.8 to 3.6) | | 2.1*** (1.9 to 2.4) | 2.8*** (2.4 to 3.3) | | | | |
| Ethnicity | | | | | | | | |
| British/English/Scottish/Welsh/Northern Irish (White) (ref) | | | | | | | | |
| Other White | | 1.5** (1.2 to 2.1) | 1.1 (0.9 to 1.4) | 0.8 (0.5 to 1.1) | 1.3 (0.6 to 1.9) | 0.8 (0.5 to 1.2) | 1.5* (1.1 to 2.1) | 1.3† (0.9 to 1.7) |
| Black, Asian and minority ethnic (BAME) | | 1.4** (1.6 to 2.4) | 0.7*** (0.5 to 0.8) | 0.7* (0.5 to 0.9) | 2.1*** (1.4 to 3.0) | 0.8 (0.6 to 1.1) | 1.1 (0.9 to 1.5) | 0.6*** (0.5 to 0.8) |
| Age group | | | | | | | | |
| 16–24 (ref) | | | | | | | | |
| 25–44 | | | 1.2 (0.9 to 1.6) | 1.2 (0.9 to 1.6) | | 1.1 (0.6 to 2.1) | | 1.2 (0.9 to 1.7) |
| 45–64 | | | 1.3† (0.9 to 1.7) | 1.3† (0.9 to 1.7) | | 1.2 (0.7 to 2.2) | | 1.3 (0.9 to 1.8) |
| 65–74 | | | 0.9 (0.7 to 1.2) | 0.9 (0.7 to 1.2) | | 1.0 (0.5 to 1.9) | | 0.8 (0.6 to 1.2) |
| 75+ | | | 0.7* (0.5 to 0.9) | 0.7* (0.5 to 0.9) | | 0.6† (0.3 to 1.1) | | 0.8 (0.5 to 1.2) |
| Highest qualification | | | | | | | | |
| No qualification (ref) | | | | | | | | |
| GCSE or lower | | | 1.1 (0.9 to 1.3) | 1.1 (0.9 to 1.3) | | 1.01 (0.8 to 1.5) | | 1.2 (0.9 to 1.4) |
| A level | | | 1.3** (1.1 to 1.7) | 1.3** (1.1 to 1.7) | | 1.5† (0.9 to 2.1) | | 1.3 (0.9 to 1.7) |
| Degree | | | 1.4*** (1.2 to 1.7) | 1.4*** (1.2 to 1.7) | | 1.5* (1.1 to 1.9) | | 1.4** (1.2 to 1.7) |
| Subjective current financial situation | | | | | | | | |
| Living comfortably (ref) | | | | | | | | |
| Doing alright | | | 1.3*** (1.2 to 1.4) | 1.3*** (1.2 to 1.4) | | 1.5*** (1.3 to 1.8) | | 1.2** (1.1 to 1.3) |
| Just about getting by | | | 2.0*** (1.8 to 2.3) | 2.0*** (1.8 to 2.3) | | 2.4*** (1.9 to 3.0) | | 1.9*** (1.6 to 2.2) |

Continued

**Table 4** Continued

| | Model A All respondents | | | | Model B Men | | Model C Women | |
|---|---|---|---|---|---|---|---|---|
| | A1 | A2 | A3 | A4 | B1 | B2 | C1 | C2 |
| Finding it quite difficult/very difficult | | | 3.8*** (3.1 to 4.6) | 3.8*** (3.1 to 4.6) | | 4.3*** (3.0 to 6.1) | | 3.6*** (2.8 to 4.5) |
| Subjective future financial situation | | | | | | | | |
| Better off (ref) | | | | | | | | |
| Worse off | | | 1.5*** (1.3to 1.8) | 1.5*** (1.3 to 1.8) | | 1.6*** (1.2 to 2.1) | | 1.5*** (1.2 to 1.7) |
| About the same | | | 0.9 (0.8 to 1.0) | 0.9 (0.8 to 1.0) | | 0.9 (0.8 to 1.2) | | 0.9 (0.8 to 1.0) |
| Live with a partner | | | | | | | | |
| No (ref) | | | | | | | | |
| Yes | | | 1.2** (1.1 to 1.4) | 1.2** (1.1 to 1.4) | | 1.3* (1.0 to 1.6) | | 1.2** (1.0 to 1.3) |
| Children in the house | | | | | | | | |
| None (ref) | | | | | | | | |
| At least one child aged 0–4 | | | 1.1 (0.9 to 1.3) | 1.1 (0.9 to 1.3) | | 1.1 (0.8 to 1.6) | | 1.1 (0.9 to 1.4) |
| At least one schoolchild aged 5–18 | | | 1.2** (1.0 to 1.3) | 1.2** (1.0 to 1.3) | | 1.2 (0.9 to 1.5) | | 1.1† (1.0 to 1.3) |
| Key worker | | | | | | | | |
| No (ref) | | | | | | | | |
| Yes | | | 1.0 (0.9 to 1.1) | 1.0 (0.9 to 1.1) | | 0.9 (0.8 to 1.1) | | 1.1 (0.9 to 1.2) |
| Not in paid or self-employed work | | | 0.8** (0.7 to 0.9) | 0.8** (0.7 to 0.9) | | 0.7** (0.6 to 1.0) | | 0.9 (0.8 to 1.0) |
| Has had symptoms that could be coronavirus | | | | | | | | |
| No (ref) | | | | | | | | |
| Yes | | | 1.2** (1.1 to 1.4) | 1.2** (1.1 to 1.4) | | 1.2 (0.9 to 1.5) | | 1.2* (1.0 to 1.4) |
| Feel lonely | | | | | | | | |
| Hardly ever (ref) | | | | | | | | |
| Sometimes | | | 3.4*** (3.2 to 3.7) | 3.4*** (3.2 to 3.7) | | 4.3*** (3.7 to 5.0) | | 3.1*** (2.8 to 3.4) |
| Often | | | 12.8*** (11.1 to 14.7) | 12.8*** (11.1 to 14.7) | | 15.5*** (11.9 to 20.2) | | 11.7*** (9.9 to 13.8) |

Continued

**Table 4** Continued

| | Model A All respondents | | | | Model B Men | | Model C Women | |
|---|---|---|---|---|---|---|---|---|
| | A1 | A2 | A3 | A4 | B1 | B2 | C1 | C2 |
| **Prior problem of sleep** | | | | | | | | |
| No (ref) | | | | | | | | |
| Yes | | | 4.5*** (3.9 to 5.1) | 4.5*** (3.9 to 5.1) | | 5.1*** (3.9 to 6.5) | | 4.1*** (3.5 to 4.8) |
| **Region** | | | | | | | | |
| North East (ref) | | | | | | | | |
| North West | | | 1.2 (0.9 to 1.6) | 1.2 (0.9 to 1.6) | | 1.0 (0.6 to 1.8) | | 1.3 (0.9 to 1.9) |
| Yorkshire and the Humber | | | 1.3† (0.9 to 1.9) | 1.3† (0.9 to 1.9) | | 1.1 (0.6 to 2.0) | | 1.5† (1.0 to 2.2) |
| East Midlands | | | 1.2 (0.9 to 1.7) | 1.2 (0.9 to 1.7) | | 1.2 (0.7 to 2.1) | | 1.3 (0.9 to 1.9) |
| West Midlands | | | 1.4* (1.0 to 1.9) | 1.4* (1.0 to 1.9) | | 1.5 (0.8 to 2.6) | | 1.4 (0.9 to 2.0) |
| East of England | | | 1.3† (0.9 to 1.8) | 1.3† (0.9 to 1.8) | | 1.5 (0.9 to 2.7) | | 1.2 (0.8 to 1.8) |
| London | | | 1.7** (1.2 to 2.4) | 1.7** (1.2 to 2.4) | | 1.6 (0.9 to 2.9) | | 1.7** (1.2 to 2.6) |
| South East | | | 1.2 (0.9 to 1.7) | 1.2 (0.9 to 1.7) | | 1.3 (0.8 to 2.3) | | 1.2 (0.8 to 1.7) |
| South West | | | 1.1 (0.8 to 1.6) | 1.1 (0.8 to 1.6) | | 1.2 (0.7 to 2.1) | | 1.1 (0.8 to 1.6) |
| Wales | | | 1.2 (0.8 to 1.6) | 1.2 (0.8 to 1.6) | | 1.3 (0.7 to 2.3) | | 1.1 (0.7 to 1.7) |
| Scotland | | | 1.2 (0.9 to 1.7) | 1.2 (0.9 to 1.7) | | 1.2 (0.7 to 2.1) | | 1.2 (0.8 to 1.8) |
| Northern Ireland | | | 1.4 (0.9 to 1.9) | 1.4 (0.9 to 1.9) | | 1.4 (0.7 to 2.6) | | 1.3 (0.9 to 2.1) |
| **Month** | | | | | | | | |
| April (ref) | | | | | | | | |
| May | | | 0.7*** (0.6 to 0.8) | 0.9 (0.8 to 1.1) | | 0.9 (0.8 to 1.1) | | 0.6*** (0.6 to 0.7) |
| June | | | 0.7*** (0.7 to 0.8) | 1.0 (0.8 to 1.2) | | 1.0 (0.8 to 1.2) | | 0.7*** (0.6 to 0.7) |
| July | | | 0.5*** (0.4 to 0.5) | 0.6*** (0.5 to 0.7) | | 0.6*** (0.5 to 0.7) | | 0.4*** (0.4 to 0.5) |
| September | | | 0.6*** (0.5 to 0.7) | 0.7*** (0.6 to 0.8) | | 0.7*** (0.5 to 0.8) | | 0.6*** (0.5 to 0.6) |
| November | | | 0.7*** (0.6 to 0.8) | 0.8† (0.7 to 1.0) | | 0.8† (0.7 to 1.0) | | 0.7*** (0.6 to 0.8) |
| January | | | 0.6*** (0.6 to 0.7) | 0.8* (0.7 to 0.9) | | 0.8** (0.6 to 0.9) | | 0.6*** (0.5 to 0.7) |
| **Interaction gender–month** | | | | | | | | |
| Women–May | | | | 0.7*** (0.5 to 0.8) | | | | |
| Women–June | | | | 0.6*** (0.5 to 0.8) | | | | |
| Women–July | | | | 0.7** (0.6 to 0.9) | | | | |
| Women–September | | | | 0.8 (0.7 to 1.1) | | | | |

Continued

**Table 4** Continued

| | Model A All respondents | | | | Model B Men | | Model C Women | |
|---|---|---|---|---|---|---|---|---|
| | A1 | A2 | A3 | A4 | B1 | B2 | C1 | C2 |
| Women–November | | | | **0.8\* (0.6 to 0.9)** | | | | |
| Women–January | | | | **0.7\*\*\* (0.6 to 0.9)** | | | | |
| Interaction ethnicity–month | | | | | | | | |
| BAME–May | | | | **1.4† (1.0 to 2.1)** | | | | |
| BAME–June | | | | 1.2 (0.8 to 1.7) | | | | |
| BAME–July | | | | 1.1 (0.7 to 1.6) | | | | |
| BAME–September | | | | **0.7† (0.5 to 1.0)** | | | | |
| BAME–November | | | | 1.0 (0.7 to 1.5) | | | | |
| BAME–January | | | | 0.9 (0.6 to 3.4) | | | | |
| Other White–May | | | | **1.6\* (1.0 to 2.7)** | | | | |
| Other White–June | | | | 1.4 (0.9 to 2.2) | | | | |
| Other White–July | | | | 1.2 (0.7 to 2.0) | | | | |
| Other White–September | | | | 1.4 (0.9 to 2.3) | | | | |
| Other White–November | | | | 1.5 (0.9 to 2.4) | | | | |
| Other White–January | | | | 1.2 (0.7 to 1.9) | | | | |
| Likelihood Ratio test versus logistic model (P value) | <0.001 | <0.001 | <0.001 | <0.001 | <0.001 | <0.001 | <0.001 | <0.001 |
| Person-month (n) | 57 141 | 57 141 | 57 141 | 57 141 | 23 926 | 23 926 | 33 215 | 33 215 |

Source: Authors' analysis, Understanding Society: COVID-19 Study, 2020.
\*\*\*P<0.001; \*\*p<0.01; \*p<0.05. Figures shown in bold are significant at least p<0.05.
†P<0.1.
BAME, Black, Asian and minority ethnic.

and Whites but the pattern is unclear for other racial/ethnic minority groups. Future studies might investigate whether some ethnic minorities are more at risk of sleep loss to aid the design of more targeted sleep interventions.

Some limitations should be acknowledged. First, sleep loss was measured by self-reports and is therefore subject to recall bias and participants' perceptions.[31] Second, attrition between data collection waves and missing values on the outcome variable meant that follow-up data were not available for more than half of participants who were surveyed in wave 1. The subsample lost to follow-up were older and less affluent than those who participated in subsequent waves. This attrition could have confounded the identification of prospective associations between gender, ethnicity and sleep loss, in that more vulnerable people were not retained in the analyses. Furthermore, loss to follow-up was slightly more likely among those who reported sleep loss at wave 1 of the COVID-19 study, which may lead to underestimated sleep loss in the study.

In conclusion, the COVID-19 pandemic and the policy responses to it, including home working and schooling, have widened the disparity of sleep deprivation across gender and ethnicity, putting women and ethnic minorities at an even greater disadvantage. Disrupted and poor sleep is associated with wider mental and physical health challenges. Policy makers and health professionals need to take action now to support and promote better sleep health among vulnerable groups during the pandemic, averting future secondary complications.

**Contributors** All authors (JCF, ME, MQ, AV) contributed equally to the initial discussion of the idea behind the manuscript, to the design of the manuscript, to the analysis plan and to the redrafting and finalisation of the manuscript. MQ conducted the statistical analysis for the paper and drafted sections of the initial draft of the manuscript. JF is responsible for the overall content as the guarantor.

**Funding** This research was supported by the Economic and Social Research Council Centre for Population Change (grant number ES/K007394/1) at the University of Southampton.

**Competing interests** None declared.

**Patient consent for publication** Not required.

**Ethics approval** This study uses secondary data for the collection of which ethical approval has been obtained by the survey team. All relevant ethical guidelines at the University of Southampton have been followed. Participants gave informed consent to participate in the study before taking part.

**Provenance and peer review** Not commissioned; externally peer reviewed.

**Data availability statement** Data are available upon reasonable request. The data are publicly available. Data from the Understanding Society Covid-19 surveys are available to download from the UK Data Service.

**ORCID iDs**
Min Qin http://orcid.org/0000-0002-5941-9979
Athina Vlachantoni http://orcid.org/0000-0003-1539-3057

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
