## [Reviewer comments · BMJ Open]

ARTICLE DETAILS

TITLE (PROVISIONAL)	A prospective longitudinal study of “Sleepless in Lockdown”: unpacking differences in sleep loss during the coronavirus pandemic in the UK
AUTHORS	Falkingham, J; Evandrou, Maria; Qin, Min; Vlachantoni, Athina

VERSION 1 – REVIEW

REVIEWER	Vézina-Im, Lydi-Anne Baylor College of Medicine, Pediatrics-Nutrition
REVIEW RETURNED	06-Jul-2021

GENERAL COMMENTS	The authors did an excellent job at integrating my previous comments and I believe that the manuscript has improved as a result. I have a few comments on elements that could further improve the article. Major Comments 1) Lines 18-19 on page 6: The authors mention that “loss to follow-up was slightly more likely amongst those who reported sleep loss at wave 1 of the COVID-19 study”. I was thus expecting that the authors would mention in their discussion that they may have underestimated sleep loss in their study. I believe it is important to add this information when discussing how the COVID-19 pandemic seems to be associated with sleep loss. 2) Lines 16-20 on page 11: The authors mention that the fact that being a member of BAME community was associated with a reduced chance of sleep loss highlights the complex relationship between ethnicity and sleep health. I would have liked if the authors had added a hypothesis to explain this somewhat surprising result. For example, maybe there were some differences in sleep between the different ethnicities included in the BAME definition? Future studies might want to investigate if some ethnic minorities are more at risk for sleep loss to design more targeted sleep interventions. I would not be surprised if some previous studies had found differences in sleep (and maybe sleep loss) based on ethnicity. For example, some cultures promote taking naps in the middle of the day (siesta) and biphasic/bimodal sleep. Minor Comments 1) Line 9 on page 5: There is a mistake. Please replace “si” by “is” (the phased end of which is scheduled for 21 June). 2) Line 44 on page 9: There is a mistake. Please replace “means” by “meals” (eating meals).
---

REVIEWER	You, Zhiying University of Colorado Denver School of Medicine, Medicine
REVIEW RETURNED	21-Jul-2021

GENERAL COMMENTS	Good to see clear description of statistics.
--

REVIEWER	Mo, Xiaokui Ohio State University, Biostatistic Center and Department of Bioinformatics
REVIEW RETURNED	18-Oct-2021

GENERAL COMMENTS	The study of this manuscript is very important, and the analytic results provide important information to governments for policy making/revisions. Research plan was well design and conducted very well. Manuscript was well-written. I have some suggestions for the authors to improve the quality of the manuscript.  1. Page 6 line 9 “the phased end of which si scheduled for 21 June.”, not sure what the si mean? 2. Table 1: Need to specify that the table 1 is for analytical sample (on table tile). Since this is analytical sample, chi-square tests and ANOVA have to be conducted for categorical and continuous variables, respectively. 3. Figure 1 is for the whole population and the sample sizes are decreasing. Some conclusions might be biased since populations across waves changed. I would suggest to make a similar figure for the analytical sample. 4. Table 2 is hard to follow, especially for the trend of sleep patterns among ethnicity groups. I would suggest to use stacking bar charts for gender and ethnicity, separately. By doing this, the changes of percent/frequencies across waves will be easily to see.
---

VERSION 1 – AUTHOR RESPONSE

Reviewer: 1

Dr. Lydi-Anne Vézina-Im, Baylor College of Medicine

Comments to the Author:

The authors did an excellent job at integrating my previous comments and I believe that the manuscript has improved as a result. I have a few comments on elements that could further improve the article.

Major Comments

1) Lines 18-19 on page 6: The authors mention that “loss to follow-up was slightly more likely amongst those who reported sleep loss at wave 1 of the COVID-19 study”. I was thus expecting that the authors would mention in their discussion that they may have underestimated sleep loss in their study. I believe it is important to add this information when discussing how the COVID-19 pandemic seems to be associated with sleep loss.

Authors' response: Thank you for this comment. We have added this point in the discussion (limitation paragraph): “Secondly, attrition between data collection waves and missing values on the outcome variable meant that follow-up data were not available for more than half of participants who were surveyed in Wave 1. The sub-sample lost to follow-up were older and less affluent than those who participated in subsequent waves. This attrition could have confounded the identification of prospective associations between gender, ethnicity and sleep loss, in that more vulnerable people

were not retained in the analyses. Furthermore, loss to follow-up was slightly more likely amongst those who reported sleep loss at wave 1 of the COVID-19 study, which may lead to underestimated sleep loss in the study.” (p.11)

2) Lines 16-20 on page 11: The authors mention that the fact that being a member of BAME community was associated with a reduced chance of sleep loss highlights the complex relationship between ethnicity and sleep health. I would have liked if the authors had added a hypothesis to explain this somewhat surprising result. For example, maybe there were some differences in sleep between the different ethnicities included in the BAME definition? Future studies might want to investigate if some ethnic minorities are more at risk for sleep loss to design more targeted sleep interventions. I would not be surprised if some previous studies had found differences in sleep (and maybe sleep loss) based on ethnicity. For example, some cultures promote taking naps in the middle of the day (siesta) and biphasic/bimodal sleep.

Authors’ response: This is an excellent suggestion. We have added the hypothesised explanation in the discussion regarding the fact that being a member of BAME community associated with a reduced chance of sleep loss highlighting the complex relationship between ethnicity and sleep health. The text reads as “One possible explanation might be that there are some differences in sleep patterns between the different ethnicities included in the BAME definition, given the vast social and cultural heterogeneity within the BAME population. Sleep is culturally practised across racial and ethnic groups²⁹. For example, some cultures in Latin and Caribbean communities promote taking naps in the middle of the day (siesta) and biphasic sleep. Others endorse the polyphasic sleep modality in Asia and Africa, anchoring their sleep at night but taking several daytime naps as needed when under a social condition that allows sleep to occur²⁹. A review study³⁰ shows that among the few published within racial/ethnic group analyses, there are differences in sleep between non-US-born and US-born racial/ethnic groups. Still, the group with the more favourable sleep profile is consistent for non-US-born Latinos compared to US-born Latinos and Whites but the pattern is unclear for other racial/ethnic minority groups. Future studies might investigate whether some ethnic minorities are more at risk of sleep loss to aid the design of more targeted sleep interventions.” (p.11)

References:

29. Williams NJ, Grandner MA, Snipes SA, Rogers A, Williams O, Airhihenbuwa C, Jean-Louis G. Racial/ethnic disparities in sleep health and health care: importance of the sociocultural context. *Sleep Health* 2015; 1: 28–35.

30. Johnson DA, Jackson CL, Williams NJ, Alcántara C. Are sleep patterns influenced by race/ethnicity – a marker of relative advantage or disadvantage? Evidence to date. *Nature and Science of Sleep* 2019; 11:79–95.

Minor Comments

1) Line 9 on page 5: There is a mistake. Please replace “si” by “is” (the phaser end of which is scheduled for 21 June).

Authors’ response: Thank you for pointing out this typo. It has been revised as “is”.

2) Line 44 on page 9: There is a mistake. Please replace “means” by “meals” (eating meals).

Authors’ response: The word “means” has been replaced as “meals”.

Reviewer: 2

Dr. Zhiying You, University of Colorado Denver School of Medicine, Colorado School of Public Health
Comments to the Author:

Good to see clear description of statistics.

Authors’ response: We are pleased with the reviewer’s comments on the statistics.

Reviewer: 3

Dr. Xiaokui Mo, Ohio State University

Comments to the Author:

The study of this manuscript is very important, and the analytic results provide important information to governments for policy making/revisions. Research plan was well design and conducted very well. Manuscript was well-written. I have some suggestions for the authors to improve the quality of the manuscript.

1. Page 6 line 9 “the phased end of which si scheduled for 21 June.”, not sure what the si mean?

Authors’ response: It has been revised as “is”.

2. Table 1: Need to specify that the table 1 is for analytical sample (on table tile). Since this is analytical sample, chi-square tests and ANOVA have to be conducted for categorical and continuous variables, respectively.

Authors’ response: Thank you for this suggestion. In the previous manuscript, we compared some selected variables and the results presented in Appendix Table 1. In this revision, we deleted Appendix Table 1. All comparisons have been presented in the revised Table 1. Chi-square and T-test were conducted for categorical and continuous variables, respectively, compared with the entire sample in Wave1. The title of Table 1 has been revised as “Analytical sample characteristics (Wave1)”. (p.13)

3. Figure 1 is for the whole population and the sample sizes are decreasing. Some conclusions might be biased since populations across waves changed. I would suggest to make a similar figure for the analytical sample.

Authors’ response: We have added a figure (Appendix Figure 1) for the prevalence of reported sleep loss pre and during the pandemic among the analytical sample. A brief description reads as “The prevalence of reported sleep loss pre and during the pandemic among the analytical sample (Appendix Figure 1) and shows a similar pattern. However, sleep loss among BAME women was lower in the analytical sample than among all respondents without follow-up selection, whilst the level of sleep loss amongst “Other White” women was higher.” (p.9; p.22)

The potential attrition effects have been discussed in the limitation paragraph. The text reads as “Secondly, attrition between data collection waves and missing values on the outcome variable meant that follow-up data were not available for more than half of participants who were surveyed in Wave 1. The sub-sample lost to follow-up were older and less affluent than those who participated in subsequent waves. This attrition could have confounded the identification of prospective associations between gender, ethnicity and sleep loss, in that more vulnerable people were not retained in the analyses. Furthermore, loss to follow-up was slightly more likely amongst those who reported sleep loss at wave 1 of the COVID-19 study, which may lead to underestimated sleep loss in the study.” (p.11)

4. Table 2 is hard to follow, especially for the trend of sleep patterns among ethnicity groups. I would suggest to use stacking bar charts for gender and ethnicity, separately. By doing this, the changes of percent/frequencies across waves will be easily to see.

Authors’ response: In line with the Reviewer’s suggestion, we have added two stacking bar charts Figure2 (a and b), illustrating the changes of percent across waves by gender and ethnicity. Relevant figure description has also added in the text: “Patterns of the changes of percent across waves are illustrated using stacked bar charts for gender and ethnicity in Figure 2a and Figure 2b.” (p.9; p.21)

VERSION 2 – REVIEW

REVIEWER	Vézina-Im, Lydi-Anne Baylor College of Medicine, Pediatrics-Nutrition
REVIEW RETURNED	19-Nov-2021

GENERAL COMMENTS	The authors did an excellent job at integrating all my comments. I believe this article will make a great contribution to the field of sleep health.
--

REVIEWER	Mo, Xiaokui Ohio State University, Biostatistic Center and Department of Bioinformatics
REVIEW RETURNED	01-Dec-2021

GENERAL COMMENTS	The authors did an excellent job at responding to my previous suggestions. They also discussed important findings in depth based on new figures. This manuscript has been improved so much!
---